# Gene Structure Evolution of the Short-Chain Dehydrogenase/Reductase (SDR) Family

**DOI:** 10.3390/genes14010110

**Published:** 2022-12-30

**Authors:** Franco Gabrielli, Marco Antinucci, Sergio Tofanelli

**Affiliations:** 1Department of Biology, University of Pisa, Via Ghini, 13-56126 Pisa, Italy; 2Department of Medicine and Life Sciences, Institute of Evolutionary Biology (UPF-CSIC), Universitat Pompeu Fabra, 08002 Barcelona, Spain

**Keywords:** SDR-family, splicing sites, orthologous genes, genetic homology, retrogenes

## Abstract

SDR (Short-chain Dehydrogenases/Reductases) are one of the oldest and heterogeneous superfamily of proteins, whose classification is problematic because of the low percent identity, even within families. To get clearer insights into SDR molecular evolution, we explored the splicing site organization of the 75 human *SDR* genes across their vertebrate and invertebrate orthologs. We found anomalous gene structures in members of the human SDR7C and SDR42E families that provide clues of retrogene properties and independent evolutionary trajectories from a common invertebrate ancestor. The same analyses revealed that the identity value between human and invertebrate non-allelic variants is not necessarily associated with the homologous gene structure. Accordingly, a revision of the SDR nomenclature is proposed by including the human *SDR40C1* and *SDR7C* gene in the same family.

## 1. Introduction

Short-chain dehydrogenase/reductase (SDR) is a superfamily of NAD(P)-dependent oxidoreductases and related enzymes, present in all living organisms [1,2,3]). For almost all members of the family, the structure of the gene, as well as the tertiary structure and the catalytic activity of the protein, have been defined. Despite their low sequence identity (typically from 10% to 30% of conserved sequence identity among families), SDR monomers have very similar tertiary structures and two specific sequences, one responsible for the specific binding of the NAD(P) coenzyme and the other including amino acids directly involved in the catalysis of various SDR substrates.

Among SDR enzymes, the divergence of the monomers increases toward their C-terminals, where the amino acid responsible for the substrate binding-site are localized [1,2,3]. Sequence variability at the SDR substrate-binding site is associated with a large spectrum of substrates, resulting in unique active sites and substrate specificities [2]. Concerning SDR differences and functions, it is worth mentioning their emerging role in human diseases. Indeed, knowledge has accumulated concerning the role that SNP variants in specific human *SDR* genes plays in a wide variety of cancers [4,5,6,7] and metabolic diseases [8,9,10,11,12]. The pattern of low sequence/high structural similarity suggests a fusion of domains specific of a common coenzyme and a wide spectrum of substrates [13]. It has also been argued that the alcohol dehydrogenase (ADH) in *Drosophila melanogaster* model organism is the ancestral “prototype” of the mammalian SDR superfamily and was responsible for the radiation of fruit fly species worldwide some 65 Mya [14]. Since then, many new data have accumulated in the genetic databases. Persson et al. [3], with an accurate analysis of protein sequences in all living species, identified 75 human SDR non-allelic variants classified, upon a clustering approach based on Hidden Markov Models (HMMs), into 48 families out of a total of 200, each having from one to eight members.

The low percent identity upon which *SDR* genes classification is based and the uncertainty about the relationships between structure and function across species make urgent a more comprehensive evolutionary analysis of *SDR* genes.

Splicing site organizations are generally conserved over very long evolutionary distances, as they demonstrated to retain information about gene homology better than other molecular traits, especially when genes belong to a large family [15,16]. Accordingly, splicing sites organization can be useful to identify cryptic homologous variants of *SDR* genes belonging either to the same or to different species [17,18]. In the present paper, we annotated the splicing site organization and the splicing site phase number of each member of the human SDR families, as well as the splicing site organization of their vertebrate and invertebrate orthologs, to achieve a more comprehensive nomenclature.

## 2. Methods

Computational tools, *SDR* gene and protein data were obtained from public primary databases: European Bioinformatics Institute (EBI), UK; the Genome Browser of the University of California, USA; National Center for Biotechnology Information (NCBI), Maryland, USA; ExPASy, Swiss Institute of Bioinformatics (SIB), Switzerland.

Vertebrate and invertebrate orthologs of human SDR variants were detected using BLASTp (release 2.13.0, NCBI). Protein sequences alignments and their identity values were obtained using Clustal Ω (release 1.2.2, EBI). Exon-intron organizations, intron physical positions, phase numbers, exon-intron boundary sequences and observance of the AG-GT/GC rule were manually checked upon gene sequence/cDNA alignments. The intron phase denotes the position of the intron within a codon: Phase 0, 1 or 2 depending whether it starts before the first base, after the first base, and after the second base, respectively. Any multiple polypeptide alignment was verified by pairwise comparison across vertebrate species; we annotated as orthologs the splicing sites having an identical phase number and identical sequence position. We used the same symbols adopted for human SDR variants by Persson et al. [3]. Symbols include SDR followed by the annotation number and a single letter: A (Atypical), C (Classical), E (Extended), U (Unknown). C and E denote the two major types of SDR family enzymes [3].

We identified the invertebrate orthologs of human SDR variants by the following procedures. First, we detected, by BLAST, the invertebrate protein having the highest sequence similarity with the human SDR protein variant. Then, we verified that the aminoacidic sequences included the *SDR* structure consensus: TGxxxGxG or TGxxGxxG and the catalytic consensus YxxxK, which are diagnostic of SDR superfamily variants [1,2,3]. SDR cofactor binding site and the catalysis active site consensus of the human SDR family are reported in the Appendix A (Online Resources 1: Appendix A). Lastly, we verified, in the invertebrate SDR variants, the presence of splicing site orthologs of those for the human variant used as BLAST probe.

Phylogenetic trees were constructed using the binarized matrix of splicing-site phases with the Wagner parsimony method, as implemented in the PARS algorithm of the software package PHYLIP version 3.6 [19]. We performed bootstrap analysis with 10,000 replications to estimate the strength of support for each clade. The same tree topologies were obtained by a Bayesian approach by BEAST 2.7.0.0 [20].

## 3. Results and Discussion

### 3.1. Gene Structure of Vertebrate and Invertebrate Variants of SDR Families

The human SDR families, classified by the relative identity values of their protein variants [3], may include members having either identical (SDR7C1 and SDR7C2), similar (SDR7C1 and SDR7C3), or completely different (SDR7C1, SDR7C4 and SDR7C5) splicing patterns (Figure 1; Table 1, Table 2 and Table 3; Online Resources 2: Appendix A; Online Resources 3: Appendix A).

Variants of seven human SDR families retain an active site made of an insertion of two amino acids: Asparagine (N) and Serine (S). The four aminoacids (N-S-Y-K) are called the catalytic tetrad [3]. We found that the human SDR families with the catalytic tetrad are SDR9C, SDR12C, SDR16C, SDR25C, SDR26C SDR28C, SDR32C (respectively Appendix A in Online Resources 2). The members of these human families have invertebrate orthologues carrying identical catalytic tetrads (Online Resources 4: Appendix A), suggesting a pre-vertebrate acquisition of these sites.

Splicing site organizations and splicing phase numbers of human SDR7C (Online Resources 3: Appendix A) and SDR21C (Online Resources 3: Appendix A) variants are identical to those of their respective orthologs in each vertebrate class. Moreover, the other SDR families present a highly conserved splicing site organization, which is identical from fishes to humans. However, a species may carry a gene, belonging to a given family, which differs with shorter or longer exons, and/or lack one splicing site, and/or bears one or two extra splicing sites with respect to its human ortholog (Online Resources 3: Appendix A). Additionally, invertebrate SDR variants may have either identical, similar, or completely different gene structure with respect to the human SDR orthologs, identified by the sequence similarity of their codified polypeptides (Online Resources 4: Appendix A). Among invertebrate proteins, we only detected a polypeptide variant (in the sea urchin *S. purpuratus*), which has an identical gene structure with human SDR11E1 and SDR11E2 variants (Online Resources 4: Appendix A), whereas variants of the other human SDR families show splicing patterns differing for the length of one or more exons, and/or by a different number of splicing sites (Online Resources 4: Appendix A).

The time of acquisition of the vertebrate gene structure can be inferred by the evolutionary position of *C. intestinalis*, a deuterostome considered a good approximation of the ancestral chordates living about 540 Mya. After the formation of the phylum, splicing patterns were conserved from fishes up to humans while genetic and protein sequences diverged (Online Resources 3: Appendix A; Online Resources 5: Appendix A).

Conversely, four human variants, SDR7C4, SDR42E1 (Online Resources 4: Appendix A), SDR21C, and SDR21C2 (Online Resources 4: Appendix A) have high identity values and completely different splicing patterns when compared to their respective invertebrate orthologs. These data show that, in SDR families, protein sequences and gene structures are phylogenetically uncoupled.

Interestingly, *Saccharomyces cerevisiae* genome has few introns, usually limited to one per gene [21]. We analyzed *S. cerevisiae* proteins having structure and catalysis consensuses diagnostic of the SDR family, but none of them had introns. Even though little evidence supports increased intron loss for paralogous gene families in plant and animal evolution [22], specific surveys on yeast suggest increased intron-loss over intron-gain events, with genes involved in metabolism, molecular transportation and enzyme activity regulation being more prone to introns loss [23]. Indeed, a recent study conducted on 263 fungal species, highlighted how the major evolutionary trend for intron changes in this kingdom, involves the loss of such sequences [24].

It is important to bear in mind, though, that analyses of Zrt-, Irt-like protein (ZIP) gene family, deemed as ancestral genes related to prion gene family evolution, revealed how intron conservation can be high when considering their relative positions in comparison of multiple sequences [25]. On the other hand, intron length conservation seems more diluted over evolutionary time scales, resulting in a heterogeneous set of intronic sequences length in ZIP genes [25].

### 3.2. Gene Structure of Human SDR7C and SDR42E Family Variants

Human SDR7C family has five non-allelic variants (Table 1; Online Resources 1: Appendix A). SDR7C1 and SDR7C2 variants have six splicing sites and an identical gene structure (Table 3). SDR7C3 variant has a very similar gene structure but only five splicing sites out of six are homologous to those of SDR7C1-2 variants (Table 3). SDR7C4 variant has only one splicing site, which has the same protein sequence position of the third splicing sites of SDR7C1, SDR7C2 and SDR7C3 variants, but a different phase number (Table 3). The SDR7C5 variant has four splicing sites and a gene structure completely different from those of all other human SDR7C family members (Table 3).

We found that human SDR40C1 variant, which is not included in the SDR7C family [3], has two splicing sites homologous to those of SDR7C1, SDR7C2 and SDR7C3 variants (Table 3), despite relatively low levels of identity with the other human SDR7C variants (26.3–31%, Table 4). Presumably, the low sequence homology prevented a correct assignment of the SDR40C1 variant into the SDR7C family by HMM-based clustering models [3]. Conversely, parsimony analyses based on the splicing-phase structure (Online resources 5: Appendix A) cluster the SDR40C1 variant within vertebrate and invertebrate orthologs of the SDR7C family with bootstrap support up to 92%. Proteins belonging to different super-families, which may have identities as low as 4% and whose homology can only be inferred from a similar 3D structure and function, may share splicing patterns [26].

### 3.3. Invertebrate Orthologs of Human SDR Family Variants

The splicing patterns of the human SDR7C family is the same in all vertebrate orthologs. Human SDR7C1, SDR7C2 and SDR7C3 variants have orthologs in several invertebrate species. *S. purpuratus* SDR7C-1C3 variant has a splicing phase identical to that of human SDR7C1 and SDR7C2 variants (Table 3). Therefore, it can be confidently considered the ancestral form. Human SDR7C4 variant has a single splicing site with zero phase number, the same protein sequence position of one splicing site of *S. purpuratus* SDR7C-1C3, *B. malayi* SDR7C-1C1 and *A. californica* SDR7C-1C3 variants. However, the invertebrate splicing sites have a different phase number (Table 3). Human SDR7C5 variant has diagnostic splicing sites and phases completely different from those of all other human SDR7C family members but shares orthologous splicing sites with invertebrate SDR7C variants. Some variants (SDR7C-1C1 and SDR7C-3C3 in *C. intestinalis*, SDR7C-2C2 in *S. purpuratus*, SDR7C-1C2 in *A. californica*) have orthologous splicing sites only with human SDR7C5 variant (Table 3), while others (SDR7C-1C2 and SDR7C-3C4 in *C. intestinalis*, SDR7C-1C1 in *S. purpuratus*) have orthologous splicing sites with human SDR7C5 as well as with human SDR7C1, SDR7C2 and SDR7C3 variants (Table 3, Online resources 5: Appendix A). We speculate that human SDR7C1, SDR7C2, SDR7C3 and SDR7C5 variants originated from a common ancestral gene which differentiated early in invertebrate evolutionary history (Online Resources 4: Appendix A).

Four splicing sites, diagnostic of human SDR40C1 variant, have orthologous splicing sites in *C. intestinalis* SDR7C-3C2 variant, which has orthologous splicing sites only with human SDR40C1 variant (Table 3). However, *S. purpuratus* SDR7C-1C3 and *A. californica* SDR7C-1C3 variants have orthologous splicing sites with human SDR40C1 as well as SDR7C1, SDR7C2 and SDR7C3 variants (Table 3). Human SDR40C1 variant shares with *C. intestinalis* SDR7C-3C2 variant a 54.57% of the sequence and below 30% with the other invertebrate orthologous variants (Table 4). Parsimony analyses confirm they are closely related (Online resources 5: Appendix A). Thus, we assume that *C. intestinalis* SDR7C-3C2 is the closest variant to the most recent common ancestor of the human SDR40C1 variant. This gene may have evolved from an invertebrate gene, an ancestor of the human gene clade SDR40C1-SDR7C1-SDR7C2-SDR7C3. These data support our hypothesis that the SDR40C1 variant is a member of the SDR7C family.

Human SDR42E family has only two non-allelic variants: SDR42E1 and SDR42E2 (Table 5), which have one and ten splicing sites, respectively (Table 6). However, SDR42E1 and SDR42E2 variants have relatively high sequence similarity (Figure 2, Table 7), with consensus regions spread in distinct blocks throughout the polypeptide molecule (Online Resources 2, Appendix A), and their gene structures are identical to those of all their respective vertebrate orthologs (Online Resources 3, Appendix A). Only the human SDR42E2 variant has orthologous splicing sites with the invertebrate SDR variants (Table 6). In particular, the ten human SDR42E2 splicing sites have orthologs in several invertebrate variants and all the ten splicing sites in *C. intestinalis* SDR42E-1E1 variant (Table 6). Such splicing pattern is also phylogenetically supported (Online resources 5: Appendix A). Thus, the *C. intestinalis* SDR42E-1E1 gene may be the closest proxy of the more recent common ancestor of invertebrate and human SDR42E variants.

### 3.4. Human SDR7C4 and SDR42E1 Are Possibly Active Retrogenes

We speculate that human *SDR7C4* and *SDR42E1* genes are active retrogenes. Retrogenes are generated from processed mRNA, do not have splicing sites and are not transcribed. However, a certain number of retrogenes are functionally active [27,28]. They may have inherited the promoter bound to the coding sequence of the parental gene or accidentally acquired a new promoter from another gene when the retrogene sequence is inserted in the DNA of a given chromosome. To be heritable, the retrotransposition needs to occur in a germline or during early embryonic stages [28]. Many active retrogenes have been discovered in mammals, and many of them developed their functional role in the germ line. After retrotransposition, active retrogenes may acquire new introns [29].

Human *SDR7C4* gene has the basic characteristics of an active retrogene. It is highly expressed in the testis (NCBI, gene ID: 57665), its chromosome localization is different from any other human SDR7C variant (Online Resources 2, Appendix A) and it has a single splicing site, which does not have orthologs with identical phase number in the invertebrate variants (unlike other SDR7C variants). Moreover, the human SDR7C4 variant has high identity values with several invertebrate variants, orthologs of other human SDR7C variants (Table 4). We interpreted these data assuming that the human SDR7C4 variant was formed by retroposition of a metazoan ancestor gene before the chordate radiation.

Likewise, the human *SDR42E1* gene has the basic characteristics of an active retrogene. *SDR42E1* gene is highly synthetized in human testis, (NCBI, gene ID: 93517) and is localized on a chromosome different from those of *SDR42E2* in vertebrate species except Catarrhines, where *SDR42E1* and *SDR42E2* are localized on the same chromosome. *SDR42E1* has a single splicing site that is shared across all vertebrate species (Online Resources 3, Appendix A) but does not have orthologs in the analyzed invertebrate variants (Table 6). Moreover, the human SDR42E1 variant shares slightly higher percent identity values with the invertebrate SDR42E2 ortholog variants than those of the human SDR42E2 variant (Table 7). These data suggest that human SDR42E1 variant has been generated by retrotransposition of an invertebrate ortholog of human SDR42E2 variant before the formation of the chordate phylum.

We could not find reported data about *SDR42E1* and *SDR42E2* chromosomal localizations in fish, amphibian, and reptilian species. However, vertebrate SDR42E1 and SDR42E2 variants have different chromosomal localization from birds up to early primate species (*Callithrix jacchus* and *Microcebus murinus*) and the same chromosomal localization in Catarrhini (Online Resources 3, Appendix A; Online Resources 6, Appendix A). We interpreted these data assuming that, during evolution, a genomic rearrangement brought *SDR42E1* and *SDR42E2* genes on the same chromosome. This hypothesis is supported by the observation that, in Catarrhines, *SDR42E1* and *SDR42E2* gene loci have identical relative positions and comparable distances whereas SDR42E1 and SDR42E2 proteins have the same molecular characteristics of those of early primates and other vertebrate species which have *SDR42E1* and *SDR42E2* genes localized on different chromosomes (Online Resources 3, Appendix A; Online Resources 6, Appendix A). These data suggest that human *SDR42E1* is an active retrogene and not a duplicated form of *SDR42E2* gene, thus adding a new case-study to stress the importance of retroposition in gene evolution [30,31,32,33,34].

## 4. Conclusions

A deeper insight into the molecular evolution of *SDR* gene families allowed us to resolve classification schemes and evolutionary patterns, regardless of the low sequence homology. The sequences of one member of the human *SDR7C* and *SDR42E* gene families retain traces of a very deep divergence time, at the root of chordate clade. The human *SDR7C4* and *SDR42E1* genes show the properties of an active retrogene, while the human *SDR40C1* gene shows a conservative splicing formula which suggests its inclusion in the same protein family of the SDRC7 variants.

## Figures and Tables

**Figure 1 genes-14-00110-f001:**
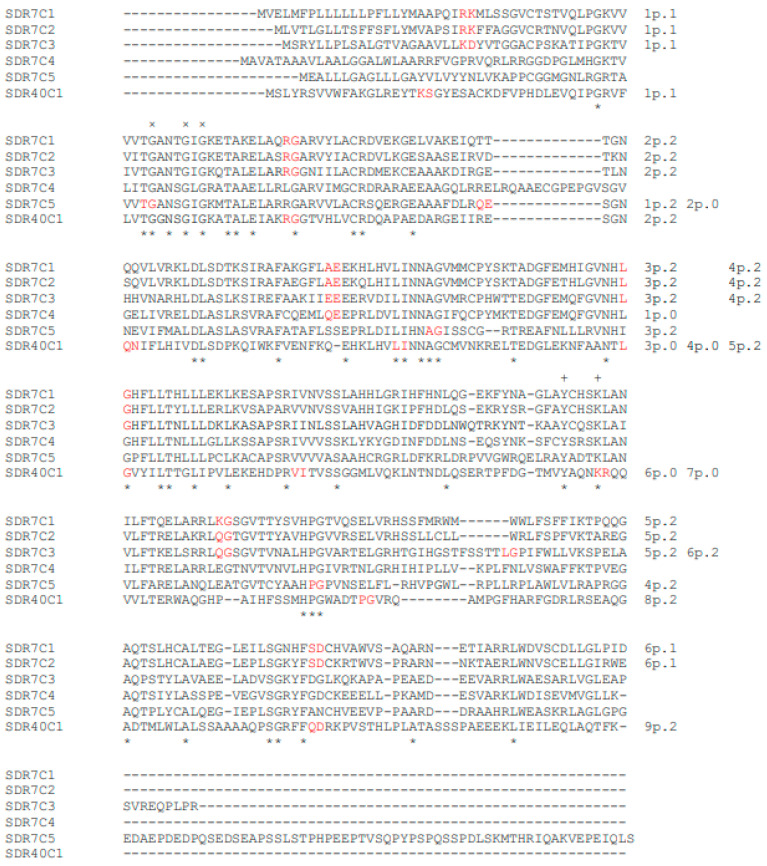
Alignment of the human SDR7C family and SDR40C1 protein variants. × and + symbols mark the structure consensus and the catalysis consensus respectively. The couples of amino acid symbols in red mark the splicing-site positions. Splicing sites are progressively numbered, and phase (p.) type is indicated after the splicing-site number. * symbol marks the position of identical amino acid residues of the aligned SDR protein sequences.

**Figure 2 genes-14-00110-f002:**
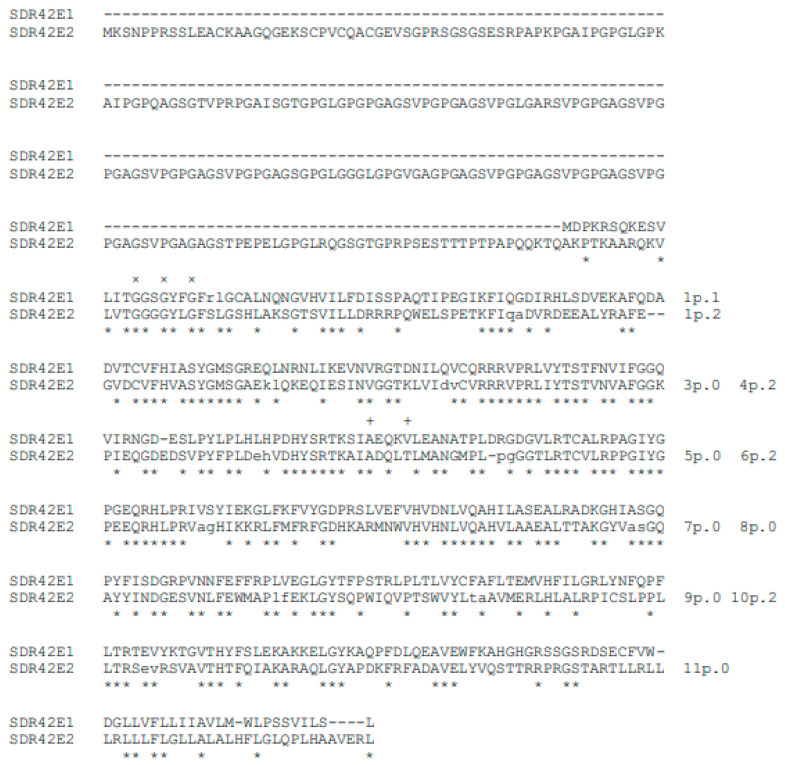
Alignment of the human SDR42E protein variants. × and + symbols mark the structure consensus and the catalysis consensus respectively. For further details, see Figure 1.

**Table 1 genes-14-00110-t001:** Genetic and molecular data of the human SDR7C family and human SDR40C1 protein variants. Chr, chromosome; the variant phase formula includes phase type symbols aligned according to the sequence of their relative splicing sites; aa n., number of the variant amino acids; standard amino acids of the consensuses are in red.

Family Symbol and Name	Enzyme Symbol	Gene Symbol	Gene ID	Chr	Exon Number	Phase Formula	aa n.	Structure Consensus	Catalysis Consensus
	SDR7C1	RDH11	51109	14	7	122221	318	GANTGIG	YCHSK
	SDR7C2	RDH12	145226				316		
SDR7C	SDR7C3	RDH13	112724	19	7	122222	331		
	SDR7C4	RDH14	57665	12	2	0	336	GANSGLG	YSRSK
Retinol dehydrogenase	SDR7C5	DHRS13	147015	17	5	2022	377	GANSGIG	YADTK
SDR40C Dehydrogenase/reductase SDR family	SDR40C1	DHRS12	79758	13	10	120020022	317	GGNSGIG	YAQNK

**Table 2 genes-14-00110-t002:** The percent identity of the human SDR7C family and human SDR40C1 protein variants.

	% Identity				
	SDR7C1	SDR7C2	SDR7C3	SDR7C4	SDR7C5
SDR7C2	71.66				
SDR7C3	49.68	48.87			
SDR7C4	46.15	46.47	48.88		
SDR7C5	45.78	46.41	42.32	44.48	
SDR40C1	32.54	33.22	30.64	31.21	28.23

**Table 3 genes-14-00110-t003:** Splicing-site organization of the human SDR7C family and SDR40C1 protein variants and their respective invertebrate orthologs. Phase symbols in a same column and highlighted in green or pink, mark the orthologous splicing sites. For other details, see Table 1.

Species	Variants	Phase Formula	Splicing-Site Phases																																				
*Homo sapiens*	**SDR7C1**	122221				** 1 **		** 2 **										** 2 **					** 2 **							** 2 **									1
	SDR7C2					1		2										2					2							2									1
	SDR7C3	122222				1		2										2					2							2			2						
	SDR7C4	0																0																					
	SDR7C5	2022					2						0								2										2								
	SDR40C1	120020022		1				2						0					0				2			0		0				2							2
*Ciona intestinalis*	SDR7C-1C1	22212					2		2												2				1						2								
	SDR7C-1C2	02021									0										2				0					2									1
	SDR7C-3C2	100022		1										0												0											2		2
	SDR7C-3C3	202021					2								0						2						0				2				1				
	SDR7C-3C4	02021									0										2				0					2									1
*Strongylocentrotus purpuratus*	SDR7C-1C1	2221						2													2									2									1
	SDR7C-1C3	122221				1		2										2					2							2									1
	SDR7C-2C2	222					2														2										2								
*Musca domestica*	SDR7C-1C3	10102				1				0							1								0					2									
*Brugia malayi*	SDR7C-1C1	2022001			2								0					2					2						0							0			1
*Aplysia californica*	SDR7C-1C2	212222					2					1										2		2							2							2	
	SDR7C-1C3	122221	1					2										2					2							2									1

**Table 4 genes-14-00110-t004:** Percent identity values of the human SDR7C family and SDR40C1 protein variants and of their invertebrate orthologs.

Species	Variants	% Identity					
*Homo* *sapiens*		SDR7C1	SDR7C2	SDR7C3	SDR7C4	SDR7C5	SDR40C1
	SDR7C2	71.66					
	SDR7C3	49.20	48.89				
	SDR7C4	45.31	46.75	48.72			
	SDR7C5H	45.87	47.21	42.01	45.39		
	SDR40C1	30.20	31.00	28.05	28.72	26.33	
*Ciona* *intestinalis*	SDR7C-1C1	37.26	38.78	34.46	36.31	35.62	26.40
	SDR7C-1C2	46.05	48.11	47.68	46.08	36.79	24.64
	SDR7C-3C2	26.00	27.48	25.25	26.51	25.83	54.57
	SDR7C-3C3	36.69	35.48	33.02	32.57	34.50	25.33
	SDR7C-3C4	46.25	46.93	44.73	46.91	39.54	23.67
*Strongylocentrotus* *purpuratus*	SDR7C-1C1	54.12	53.76	53.36	50.53	44.04	30.71
	SDR7C-1C3	47.17	47.78	59.09	48.90	40.30	29.61
	SDR7C-2C2	42.49	42.77	42.86	42.44	45.02	27.63
*Musca domestica*	SDR7C-1C3	49.37	51.91	53.92	47.78	44.52	31.25
*Brugia malayi*	SDR7C-1C1	36.77	39.48	38.54	41.40	33.66	27.67
*Aplysia* *californica*	SDR7C-1C2	39.81	41.04	41.38	36.86	33.13	29.24
	SDR7C-1C3	49.05	46.47	52.05	47.34	40.26	28.57

**Table 5 genes-14-00110-t005:** Genetic and molecular data of the human SDR42E family variants. For further details, see Table 1.

Family Symbol and Name	Enzyme Symbol	Gene Symbol	Gene ID	Chr	Exons	Phase Formula	aa n.	% Identity	Structure Consensus	Catalysis Consensus
SDR42E 3-β-HSD family	SDR42E1	SDR42E1	93517	16	2	1	393	47.18	GGSGYFG	YSRTK
	SDR42E2	SDR42E2	100288072		12	20020200020	626		GGGGYLG	

**Table 6 genes-14-00110-t006:** Splicing-site organization of the human SDR42E family variants and their invertebrate orthologs. For further details see Table 1.

Species	Variants	Phase Formula	Splicing-Site Phases															
** *H. sapiens* **	**SDR42E1**	**1**			**1**													
	SDR42E2	0020200020				0	0	2	0	2	0	0		0	2	0		
*C. intestinalis*	SDR42-1E1	100202000202		1		0	0	2	0	2	0	0		0	2	0		2
*S. purpuratus*	SDR42-1E1	02020020					0	2	0	2	0	0			2	0		
*Caenorhabditis elegans*	SDR42-1E1	20000	2			0	0		0		0						0	
	SDR42-2E1	000000				0	0		0		0		0				0	
*Caenorhabditis remanei*	SDR42-1E2	00000					0		0		0		0				0	
*A. californica*	SDR42-1E1	2000020						2	0		0	0		0	2	0		

**Table 7 genes-14-00110-t007:** Percent identity values of the human SDR42E family variants and of their invertebrate orthologs.

Species	Variants	% Identity	
** *H sapiens* **		SDR42E1	SDR42E2
	SDR42E2	47.18	
*C. intestinalis*	SDR42-1E1	41.58	40.69
*S. purpuratus*	SDR42-1E1	49.55	43.88
*C. elegans*	SDR42-1E1	27.01	27.54
	SDR42-2E1	24.72	29.46
*C. remanei*	SDR42-1E2	28.33	31.58
*A. californica*	SDR42-1E1	48.97	39.69

## Data Availability

Not applicable.

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
