# Peer review of "Gene Structure Evolution of the Short-Chain Dehydrogenase/Reductase (SDR) Family"

_genes, 2022, doi:10.3390/genes14010110_

Round 1

Reviewer 1 Report

In this study, the authors study the molecular evolution of SDR, a heterogeneous superfamily of proteins whose classification is questionable due to their high heterogeneity even within families. To resolve this issue, they proposed using the splicing site organization. They do that across 75 human SDR genes across their vertebrate and invertebrate orthologs (found using BLAST). Unlike their heterogeneity in sequence, SDR monomers have very similar tertiary structures and two specific sequences that aid in their identification. The authors reported “anomalous” gene structures in members of the human SDR7C and SDR42E families and concluded the evolution of the families.

Major comments

1. Can the authors add a diagram of the SDRs? So that we can better understand the problem they faced and the problem they aim to solve.

2. In looking for SDR orthologous, did you have any negative controls?

3. line 215 summarizes a long paragraph (198-215) and refers us to Online Resources 3, which doesn’t have some of the genomes discussed above. Can you add a figure relevant to this paragraph?

4. lines 222-223, how was that calculated?

5. The author claim that they address an ongoing debate, but I see no references for opposing views. Is there a debate or is this the first argument about this new issue? As far as I know, the debate goes back 30 years ago https://www.ncbi.nlm.nih.gov/pmc/articles/PMC319428/, which means that the authors should make an effort to cite the relevant literature in one paragraph and reflect on the literature in their discussion.

6. Explain the importance of this family, funciton, and why is it interesting.

7. Can you add a diagram that summarizes your findings? i.e., construction of the evolution of this family? It’s hard to “dig” the stories in your results+discussion section (which I am not fond of). Adding such a diagram would be most helpful.

Minor comments

1. Please check your English. For example:

Line 29. Structure -> structures

Line 135: belonging a given family -> belonging to a given family

2. Why is the Strongylocentrotus purpuratus genome not cited https://doi.org/10.1126/science.1133609?

Author Response

Rev#1 pt 1. Can the authors add a diagram of the SDRs? So that we can better understand the problem they faced and the problem they aim to solve.

Rev#1 pt 7. Can you add a diagram that summarizes your findings? i.e., construction of the evolution of this family? It’s hard to “dig” the stories in your results+discussion section (which I am not fond of). Adding such a diagram would be most helpful.

We have carefully considered the suggestion of rev#1 at points 1 and 7 (and Rev#2 at point 1, see below). However, we concluded that a diagram with 75 gene families would be too space-consuming and disproportionate compared to the utility for the reader. The only change respect to previous classifications is that SDR7C and SDR40C1 should be included in the same family upon the similarity of their splicing sites. That is already shown at Tab. 2a and in Online Resources 4, Fig.25 and Table 27.

As regards the evolutionary patterns of the SDR superfamily, we retain that a single phylogram would be too complex and the relationships among subfamilies too uncertain to be easily caught by the readers. The phylograms of the relevant subfamilies are shown separately in Online Resource 6, Figs 39-49.

Rev#1 pt 2. In looking for SDR orthologous, did you have any negative controls?

We did not explicitly use a negative control. However, its use was unnecessary since the BLAST search yielded no entries other than SDR sequences. The use of ADH or other medium-chain dehydrogenases as a negative control have given the same results.

Rev#1 pt 3. line 215 summarizes a long paragraph (198-215) and refers us to Online Resources 3, which doesn’t have some of the genomes discussed above. Can you add a figure relevant to this paragraph?

The relevant figure is in Online Resources 4: Fig. 25. We changed accordingly.

Rev#1 pt 4. lines 222-223, how was that calculated?

Phylogenetic analyses are described in the Methods section lines 99-104 (last version).

Rev#1 pt 5. The author claim that they address an ongoing debate, but I see no references for opposing views. Is there a debate or is this the first argument about this new issue? As far as I know, the debate goes back 30 years ago https://www.ncbi.nlm.nih.gov/pmc/articles/PMC319428/, which means that the authors should make an effort to cite the relevant literature in one paragraph and reflect on the literature in their discussion.

We implemented Reviewer’s suggestion. We toned down the level of the controversy adding a new paragraph. Here, we introduced the seminal paper by Ladenstein et al. (2008) where the evolutionary implications about the origin of the short-chain deydrogenases were first put forward: “It has been also argued that the alcohol dehydrogenase (ADH) in Drosophila melanogaster model organism is the ancestral “prototype” of the mammalian SDR superfamily and was responsible of the radiation of fruit fly species worldwide some 65 Mya [14]. Since then, many new data have accumulated in the genetic databases.”

Rev#1 pt 6. Explain the importance of this family, funciton, and why is it interesting.

In the Introduction we emphasized the importance of the SDR family in terms of antiquity (“present in all living organisms”, line 27 last version), knowledge (“For almost all members of the family, the structure of the gene, as well as the tertiary structure and the catalytic activity of the protein, have been defined.”, lines 29-30 last version), human health (“knowledge has accumulated about the role that SNP variants in specific human SDR enzyme genes plays in a wide variety of cancers [4-7] and other metabolic diseases [8-12], lines 42-44 last version).

Minor comments

Rev#1 pt 1. Please check your English. For example:

Line 29. Structure -> structures

Line 135: belonging a given family -> belonging to a given family

Done. The text has been revised by a mother-tongue with expertise in the field.

Rev#1 pt 2. Why is the Strongylocentrotus purpuratus genome not cited https://doi.org/10.1126/science.1133609?

The sequences of S. purpuratus, like those of all other organisms, were downloaded from the databases. The bibliographic source is not reported in the text. Specifically, the sequences of S. purpuratus are from:

Tu Q, et al. Gene structure in the sea urchin Strongylocentrotus purpuratus based on transcriptome analysis. Genome Res 2012

Sea Urchin Genome Sequencing Consortium., et al. The genome of the sea urchin Strongylocentrotus purpuratus. Science 2006

Reviewer 2 Report

Gabrielli et al. report a study on the molecular evolution of the SDR gene family, which allows to resolve classification schemes and evolutionary patterns regardless of the low sequence homology. The results are interesting to the broad readership of this journal, but the paper needs minor revision before accepting for publication. 

(1)    In the Introduction, the authors are suggested to introduce the differences and functions of SDR in a clear manner. 

(2)    More importantly, the authors should introduce more about the splicing-site phases. 

(3)    The organization of tables is a bit weird. For example, Table 1b may be better renamed as Table 1 and Table 2.

(4)    The Table and its captions need to be checked carefully. For example, the caption of Table 2a indicates that there is a highlighted section, but it is not shown in the Table 2a. Besides, in line 291 of the article, the commentary is misleading. Table 26 is not available in the Online Resources 4. Please check and correct. 

(5)    Line 34: “C-terminal” should be “C-termina”, “amino-acid” should be “amino acids”.

Author Response

Rev#1 pt 1. Can the authors add a diagram of the SDRs? So that we can better understand the problem they faced and the problem they aim to solve.

Rev#1 pt 7. Can you add a diagram that summarizes your findings? i.e., construction of the evolution of this family? It’s hard to “dig” the stories in your results+discussion section (which I am not fond of). Adding such a diagram would be most helpful.

We have carefully considered the suggestion of rev#1 at points 1 and 7 (and Rev#2 at point 1, see below). However, we concluded that a diagram with 75 gene families would be too space-consuming and disproportionate compared to the utility for the reader. The only change respect to previous classifications is that SDR7C and SDR40C1 should be included in the same family upon the similarity of their splicing sites. That is already shown at Tab. 2a and in Online Resources 4, Fig.25 and Table 27.

As regards the evolutionary patterns of the SDR superfamily, we retain that a single phylogram would be too complex and the relationships among subfamilies too uncertain to be easily caught by the readers. The phylograms of the relevant subfamilies are shown separately in Online Resource 6, Figs 39-49.

Rev#1 pt 2. In looking for SDR orthologous, did you have any negative controls?

We did not explicitly use a negative control. However, its use was unnecessary since the BLAST search yielded no entries other than SDR sequences. The use of ADH or other medium-chain dehydrogenases as a negative control have given the same results.

Rev#1 pt 3. line 215 summarizes a long paragraph (198-215) and refers us to Online Resources 3, which doesn’t have some of the genomes discussed above. Can you add a figure relevant to this paragraph?

The relevant figure is in Online Resources 4: Fig. 25. We changed accordingly.

Rev#1 pt 4. lines 222-223, how was that calculated?

Phylogenetic analyses are described in the Methods section lines 99-104 (last version).

Rev#1 pt 5. The author claim that they address an ongoing debate, but I see no references for opposing views. Is there a debate or is this the first argument about this new issue? As far as I know, the debate goes back 30 years ago https://www.ncbi.nlm.nih.gov/pmc/articles/PMC319428/, which means that the authors should make an effort to cite the relevant literature in one paragraph and reflect on the literature in their discussion.

We implemented Reviewer’s suggestion. We toned down the level of the controversy adding a new paragraph. Here, we introduced the seminal paper by Ladenstein et al. (2008) where the evolutionary implications about the origin of the short-chain deydrogenases were first put forward: “It has been also argued that the alcohol dehydrogenase (ADH) in Drosophila melanogaster model organism is the ancestral “prototype” of the mammalian SDR superfamily and was responsible of the radiation of fruit fly species worldwide some 65 Mya [14]. Since then, many new data have accumulated in the genetic databases.”

Rev#1 pt 6. Explain the importance of this family, funciton, and why is it interesting.

In the Introduction we emphasized the importance of the SDR family in terms of antiquity (“present in all living organisms”, line 27 last version), knowledge (“For almost all members of the family, the structure of the gene, as well as the tertiary structure and the catalytic activity of the protein, have been defined.”, lines 29-30 last version), human health (“knowledge has accumulated about the role that SNP variants in specific human SDR enzyme genes plays in a wide variety of cancers [4-7] and other metabolic diseases [8-12], lines 42-44 last version).

Minor comments

Rev#1 pt 1. Please check your English. For example:

Line 29. Structure -> structures

Line 135: belonging a given family -> belonging to a given family

Done. The text has been revised by a mother-tongue with expertise in the field.

Rev#1 pt 2. Why is the Strongylocentrotus purpuratus genome not cited https://doi.org/10.1126/science.1133609?

The sequences of S. purpuratus, like those of all other organisms, were downloaded from the databases. The bibliographic source is not reported in the text. Specifically, the sequences of S. purpuratus are from:

Tu Q, et al. Gene structure in the sea urchin Strongylocentrotus purpuratus based on transcriptome analysis. Genome Res 2012

Sea Urchin Genome Sequencing Consortium., et al. The genome of the sea urchin Strongylocentrotus purpuratus. Science 2006

Rev#2 pt 1 In the Introduction, the authors are suggested to introduce the differences and functions of SDR in a clear manner. 

See pt 1,6,7 Rev#1.

Rev#1 pt 2 More importantly, the authors should introduce more about the splicing-site phases. 

We explicated it in the Methods section (lines 80-82 last version): “Intron phase denotes the position of the intron within a codon: phase 0, 1 or 2 depending on whether it starts before the first base, after the first base, and after the second base, respectively.”

(3)    The organization of tables is a bit weird. For example, Table 1b may be better renamed as Table 1 and Table 2.

Done. The Tables were renamed and controlled in double-check.

(4)    The Table and its captions need to be checked carefully. For example, the caption of Table 2a indicates that there is a highlighted section, but it is not shown in the Table 2a. Besides, in line 291 of the article, the commentary is misleading. Table 26 is not available in the Online Resources 4. Please check and correct. 

We corrected all the captions and the references to tables and figures in the text.  Highlighted characters are not visible because the tables have not been printed in color. For safety, we attached an xls file.

(5)    Line 34: “C-terminal” should be “C-termina”, “amino-acid” should be “amino acids”.

We checked and corrected.

Round 2

Reviewer 1 Report

Thank you for addressing my questions, I approve the publication of this piece.

Reviewer 2 Report

The manuscript has been improved after revision. But one issue remained with the caption of Fig. 2a. The authors stated that "Phase symbols in a same column and highlighted in green or pink, mark the orthologous splicing sites", but no such highlighted symbols were found in the figure.